# Quantifying transboundary transport flux of CO over the Tibetan Plateau: variabilities and drivers

Zhenda Sun [1], Hao Yin [2], Zhongfeng Pan[3], Chongyang Li[1], Xiao Lu [4], Youwen Sun [1,5]* and Cheng Liu [6,7,8,9]*

(1 *School of Environmental Science and Optoelectronic Technology, University of Science and Technology of China, Hefei 230026, China*)

(2 *School of Energy and Environment, City University of Hong Kong, Hong Kong SAR, China*)

(3 *Institutes of Physical Science and Information Technology, Anhui University, Hefei 230601, China*)

(4 *School of Atmospheric Sciences, Sun Yat-Sen University, Zhuhai 519082, China*)

(5 *Key Laboratory of Environmental Optics and Technology, Anhui Institute of Optics and Fine Mechanics, HFIPS, Chinese Academy of Sciences, Hefei 230031, China* )

(6 *Department of Precision Machinery and Precision Instrumentation, University of Science and Technology of China, Hefei 230026, China*)

(7 *Center for Excellence in Regional Atmospheric Environment, Institute of Urban Environment, Chinese Academy of Sciences, Xiamen, 361021, China*)

(8 *Key Laboratory of Precision Scientific Instrumentation of Anhui Higher Education Institutes, University of Science and Technology of China, Hefei, 230026, China*)

(9 *Anhui Province Key Laboratory of Polar Environment and Global Change, USTC, Hefei, 230026, China*)

Correspondence: ywsun@aiofm.ac.cn; chliu81@ustc.edu.cn

**Abstract:**

The Tibetan Plateau significantly impacts regional and global climate systems due to its unique geographical location and complex environmental processes. This study investigates the variability and driving force of transboundary transport flux of carbon monoxide (CO) over the Tibetan Plateau from May 2018 to April 2024. The CO fluxes were calculated with a closed-loop integral method using the TROPOMI, ERA5, and GEOS-CF data products. The results show that the external influx and internal efflux of CO over the Tibetan Plateau in each year are relatively close and have similar seasonal characteristics. High levels of CO flux occur in late autumn to winter, and low levels occur in summer. In most cases, CO flux maximizes in November, December or January, and minimizes in July or August. The month to month variability during late autumn to winter is greater than that in summer. The Tibetan Plateau has experienced an increase of 2.86 Tg CO yr$^{-1}$ in external influx, while the internal efflux has slightly decreased by -1.70 Tg CO yr$^{-1}$. The magnitude of the increase in external influx in the southwestern segment is greater than in the northeastern segment. Conversely, the magnitude of the decrease in internal efflux in the northeastern segment is greater than in the southwestern segment. The source attribution results reveal that the external input of CO into the Tibetan Plateau mainly comes from South Asia.

The increase in external influx of CO in recent years over the Tibetan Plateau are potentially
linked to the rapid rise in CO concentrations from South Asia.

**1 Introduction**

The Tibetan Plateau, often referred to as the "Third Pole of the Earth", is characterized by its
extensive snow, glaciers, permafrost, and seasonal frozen ground. Due to the complex interactions
among atmospheric, cryospheric, hydrological, geological, and environmental processes, this
region profoundly impacts global climate and water cycle systems. The Tibetan Plateau plays a
crucial role in the global climate system and serves as a critical indicator of regional and global
climate change (Qian et al., 2011; Li et al., 2017; Bibi et al., 2018; Gao et al., 2019). This plateau
and its surrounding regions experience various atmospheric circulation patterns, including the
Indian summer monsoon, winter westerlies, and the East Asian monsoon (Yao et al., 2012; Chen
et al., 2020). These circulation patterns are essential in shaping regional climate and pollutant
transport. In particular, the southwestern Tibetan Plateau is influenced by the South Asian
monsoon. During summer, the intensified monsoon transports warm, moist air masses and
pollutants, such as aerosols and particulate matter, from the South Asian subcontinent to the
Tibetan Plateau. Owing to the uplift effect of the plateau's topography, these airflows descend into
the southern and southwestern regions after crossing the Himalayas, delivering rainfall and
accumulating pollutants. Conversely, the northeastern Tibetan Plateau are impacted by the East
Asian monsoon. In winter, cold air from Central and Northern Asia flows into the plateau,
carrying pollutants from these areas. In summer, the East Asian monsoon brings moist air and
pollutants from the East Asian coast into the eastern Tibetan Plateau, affecting local air quality
(Ramanathan et al., 2005; Xu et al., 2009; Kaspari et al., 2011; Qian et al., 2011; Lüthi et al., 2015;
Cong et al., 2015; Zhang et al., 2015; Wang et al., 2019; Wu et al., 2020; Li et al., 2021; Sun et al.,
2021). Surrounding the Tibetan Plateau, densely populated Eurasian countries are experiencing
rapid economic development, leading to increased emissions of air pollutants. Consequently, these
regions have become some of the most polluted areas globally, and the pollutants can be
transported over long distances to the Tibetan Plateau via the Asian monsoon and westerly
circulation (Lawrence and Lelieveld, 2010). In addition to the rising industrial and agricultural
emissions within the Tibetan Plateau, these external pollutants can also significantly impact its
ecosystem and climate (Ji et al., 2015; Sun et al., 2021; Yin et al., 2022).

Carbon monoxide (CO) is one of the most significant atmospheric pollutants, primarily
resulting from incomplete combustion of fossil fuels, biomass burning, and the oxidation of
methane and non-methane hydrocarbons. This pollutant can indirectly exacerbate global warming
by participating in the formation and reactions of other greenhouse gases in the atmosphere. CO is
predominantly removed from the atmosphere through reactions with hydroxyl radicals (OH)
(Holloway et al., 2000; Heald et al., 2003; Luo et al., 2013; Martínez-Alonso et al., 2020). With a
lifetime ranging from several weeks to months, CO can persist in the atmosphere for extended
periods, undergoing both horizontal and vertical transport. Consequently, CO is frequently
employed as a tracer for studying pollutant transport dynamics (Holloway et al., 2000;
Gloudemans et al., 2006; Jeong and Hong, 2021). Given its unique chemical properties and
significant climatic effects, studying CO flux, variability, and driving factors offers valuable
insights into the atmospheric conditions over the Tibetan Plateau. Furthermore, CO sources may

vary across different regions, understanding their contributions to the variability of CO flux over the Tibetan Plateau is essential for developing effective management strategies.

To investigate transboundary transport flux of CO over the Tibetan Plateau, this study employs a comprehensive array of data products, including the TROPOspheric Monitoring Instrument (TROPOMI), the fifth-generation European Centre for Medium-Range Weather Forecasts Reanalysis (ERA5), and the Goddard Earth Observing System Composition Forecast (GEOS-CF). By integrating a closed-loop integral method with a regression model, we elaborate a top-down approach based on satellite remote sensing to estimate the CO flux across the Tibetan Plateau. This quantitative analysis encompasses transport flux of CO along the Tibetan Plateau boundary (hereafter the closed-loop flux), spanning from the surface at 1000 hPa to the stratosphere at 50 hPa, over a nearly six-year time series from May 2018 to April 2024. Our analysis emphasizes seasonal and inter-annual variabilities of the closed-loop CO flux. We further split the closed-loop CO flux into sub-fluxes arising from southwestern and northeastern segments, enabling us to quantify the impacts of surrounding areas on the Tibetan Plateau. This study aims to elucidate the spatiotemporal variabilities and the driving forces of the CO transport flux over the Tibetan Plateau, thereby providing scientific evidence for a deeper understanding of the Tibetan Plateau's role in global climate and environmental dynamics.

In the subsequent section, we present an overview of the Tibetan Plateau territory, providing a concise description of the dataset utilized, along with the methodologies employed for the closed-loop integral calculation and trend regression model. The third section delves into the spatiotemporal dynamics of the CO total column and transport flux over the Tibetan Plateau, highlighting the trend fitting outcomes. The fourth section elucidates the driving forces behind the variability of transboundary transport CO flux. The study culminates in the fifth section with a synthesis of our findings and conclusions.

## 2 Methodology and dataset

This section introduces geographical description of the Tibetan Plateau, the closed-loop integral method used to derive external influx and internal efflux of CO, the regression model used for trend analysis and the dataset involved. Figure 1 provides a visual representation of the specific steps involved in applying different data products for the closed-loop integral flux calculation and regression model fitting, with further details elaborated in Sections 2.2 to 2.5.

### 2.1 Geographical description of the Tibetan Plateau

We have conducted an analysis of both the external influx and internal efflux of CO across the Tibetan Plateau. The external influx refers to the quantity of CO that is transported from regions outside the Tibetan Plateau into its boundaries. Conversely, the internal efflux denotes the amount of CO that is generated within the Tibetan Plateau and dispersed to areas beyond its confines. As shown in Figure 2, we bifurcated the Tibetan Plateau's geographical boundary into two segments. The first segment encompasses the southwestern Tibetan Plateau, which is significantly influenced by the South Asian monsoon. The substantial topographic rise in this area intensifies the monsoon's impact (Huang et al., 2023). The second segment is situated in the northeastern Tibetan Plateau, where the westerlies and the East Asian monsoon play a pivotal role. There is a marked geographical disparity between these two segments.

For comparison, we annotate the adjacent regions around the Tibetan Plateau. We categorize these regions into three broad zones: the western, central, and eastern zones. The western zone predominantly comprises Himachal Pradesh, Uttarakhand, and Ladakh. The central zone is characterized by Nepal, Sikkim, and Bhutan, while the eastern zone includes Assam, Sagaing, and Kachin State, among others. A visual representation of these regional demarcations is presented in Figure 2.

## 2.2 Dataset description

We utilize a comprehensive dataset encompassing the total column of CO measured by the TROPOMI instrument, the vertical profile of CO from the GEOS-CF system, meteorological data extracted from the ERA5 reanalysis, and atmospheric air-mass simulated by the GEOS-Chem model.

TROPOMI is a push broom imaging spectrometer on the ESA Sentinel-5 platform, providing daily global coverage of CO total column at 13:30 local time (LT) (Veefkind et al., 2012; Landgraf et al., 2020). We used TROPOMI Level 2 CO and filtered the TROPOMI data according to the method of Landgraf et al. (2020), i.e., we removed all pixels with a TROPOMI quality mark below 0.5, leaving only data with no clouds or only low-altitude clouds. For convenience of calculation, we resampled the CO data product in space and time to match the spatiotemporal resolution of the meteorological field.

ERA5 is the fifth generation of atmospheric reanalysis dataset for global climate from ECMWF (European Centre for Medium-Range Weather Forecasts). It provides hourly global atmospheric, land surface and ocean wave estimates since 1950 and is produced by the Copernicus Climate Change Service (C3S) of ECMWF (Hersbach et al., 2020). We extracted the wind vectors that coincident with the Sentinel-5's overpass time with a vertical resolution of 50 hPa from 50 hPa to 900 hPa and 25 hPa from 900 hPa to 1000 hPa.

Since measurement-based CO profile is not available, the vertical CO profile from the GEOS-CF system is used to correct vertically non-uniform distribution of CO concentration and wind field. The GEOS-CF system is a near-real-time high resolution (0.25° × 0.25°) global 3D coupled chemical and meteorological modeling system developed by NASA's Global Modeling and Assimilation Office (GMAO) (Keller et al., 2021). Since 2018, GEOS-CF has provided global CO vertical profiles at 23 pressure levels (from 1000 to 10 hPa) on an hourly basis.

The air-mass dataset used for calculating CO flux comes from the simulations of GEOS-Chem model version 12.2.1 (https://doi.org/10.5281/zenodo.2580198, International GEOS-Chem Community, 2019) (Long et al., 2015). The model is driven by assimilated meteorological data obtained from the Goddard Earth Observing System (GEOS) of NASA's Global Modeling and Assimilation Office (GMAO) (Bey et al., 2001). The GEOS-FP dataset has a horizontal resolution of 0.25 ° latitude × 0.3125 ° longitude and includes 47 vertical levels from the surface to 0.01 hPa. Surface meteorological variables and planetary boundary layer height (PBLH) are provided with a 1-hour interval. We used a nested grid version of GEOS-Chem with a horizontal resolution covering the East Asia region (70–140 ° E, 15–55 ° N), with boundary conditions derived from a global simulation at a resolution of 2 ° latitude × 2.5 ° longitude (Lee and Park, 2022).

## 2.3 The closed-loop integral method for CO flux calculation

We use the closed-loop integral method from Shaiganfar et al. (2017) to calculate the CO flux along the Tibetan Plateau boundary. With this closed-loop integral method, the external influx ($Flux, in$) and internal efflux ($Flux, out$) of CO across the Tibetan Plateau can be calculated as equations (1) and (2), respectively. The calculation flow is shown in Figure 1, and the calculation methodology diagram is in Figure 2.

$$Flux, in \approx - \sum VCD(S_i) \cdot \omega_i \cdot \cos \beta_i \cdot \Delta S_i \ , \ \beta_i > 90° \quad (1)$$

$$Flux, out \approx \sum VCD(S_i) \cdot \omega_i \cdot \cos \beta_i \cdot \Delta S_i \ , \ \beta_i < 90° \quad (2)$$

$$Flux, net = Flux, in - Flux, out \quad (3)$$

Where $VCD(S_i)$ represents CO total column located at the path $S_i$, which represents the $i^{th}$ segment of the path $S$. $\beta_i$ is the angle between the wind field vector ω and the boundary normal vector $n$. If $\beta_i > 90°$, it indicates an external influx, representing the quantity of CO that is transported from regions outside the Tibetan Plateau into its boundaries. if $\beta_i < 90°$, it indicates an internal efflux, representing the amount of CO that is generated within the Tibetan Plateau but dispersed to areas beyond its confines. $\Delta S_i$ represents the integration step size. *Flux, net* calculated as the difference between *Flux,in* and *Flux,out* represent the net CO flux across the Tibetan Plateau boundary. If $Flux, net > 0$, it means that the portion transported from outside regions into Tibetan Plateau is larger than the portion transported from Tibetan Plateau to the outside regions, and vice versa for $Flux, net < 0$.

**2.4 Wind field correction and uncertainty**

ERA5 provides wind field components $u$ and $v$ along with their uncertainties $\sigma_u$ and $\sigma_v$. Therefore, the wind speed $\omega_{spd}$ and wind direction $w_{dir}$ can be calculated as equation (4),

$$\omega_{spd} = \sqrt{u^2 + v^2} , \ w_{dir} = 180 + \frac{180}{\pi} atan2(u, v) \quad (4)$$

The uncertainty in the wind field can be calculated using the error propagation formulas (5) and (6),

$$\sigma_{wspd} = \sqrt{\sum_i \left[ \left( \frac{\partial w_{spd(zi)}}{\partial u(zi)} \times \sigma_{u(zi)} \right)^2 + \left( \frac{\partial w_{spd(zi)}}{\partial v(zi)} \times \sigma_{v(zi)} \right)^2 \right]} \quad (5)$$

$$\sigma_{wdir} = \sqrt{\sum_i \left[ \left( \left( \frac{\partial w_{dir(zi)}}{\partial u(zi)} \times \sigma_{u(zi)} \right)^2 + \left( \frac{\partial w_{dir(zi)}}{\partial v(zi)} \times \sigma_{v(zi)} \right)^2 \right] \right)} \quad (6)$$

Where $\sigma_{wspd}$ and $\sigma_{wdir}$ are the uncertainties in wind speed and wind direction, respectively. $\sigma_{u(zi)}$ and $\sigma_{v(zi)}$ are the uncertainties in $u(zi)$ and $v(zi)$, $\frac{\partial w_{spd(zi)}}{\partial u(zi)}$ and $\frac{\partial w_{spd(zi)}}{\partial v(zi)}$ are the partial derivatives of $w_{spd(zi)}$ with respect to $u(zi)$ and $v(zi)$, $\frac{\partial w_{dir(zi)}}{\partial u(zi)}$ and $\frac{\partial w_{dir(zi)}}{\partial v(zi)}$ are the partial derivatives of $w_{dir}(zi)$ with respect to $u(zi)$ and $v(zi)$, respectively. The $zi$ represents the height of the wind field.

Since CO concentration and wind field are distributed non-uniformly along with the vertical height, a vertically averaged wind field is needed for flux calculation to minimize errors caused by these non-uniformities. In order to do so, we first convert the volume mixing ratio (VMR) of CO at each altitude into mass concentration (the product of vertical profile and atmospheric air mass) via equation (7). We then take it as the weighting function to correct the original wind field via equation (8) (Shaiganfar et al., 2017; Huang et al., 2020). Meanwhile, we calculate the uncertainty in the wind field following the method of Huang et al. (2020).

$$\tau_j(z_j) = \frac{x_j(z_j) \cdot Airmass_j(z_j)}{\sum_i x_j(z_j) \cdot Airmass_j(z_j)} \tag{7}$$

$$\omega_{j,avg} = \sum_i \omega(z_i) \cdot \tau_j(z_j), \quad \theta_{j,avg} = \sum_i \theta(z_i) \cdot \tau_j(z_j) \tag{8}$$

$$\sigma_{\omega z} = \sqrt{\sum_i [\tau_j(\omega_{j,avg} - \omega(z_i))]^2}, \sigma_{\theta z} = \sqrt{\sum_i [\tau_j(\theta_{j,avg} - \theta(z_i))]^2} \tag{9}$$

Where $\omega(z_i)$, $\theta(z_i)$, $x_j(z_j)$, and $Airmass_j(z_j)$ represent the wind field vector, the wind direction, CO VMR concentration, and air mass at height $z_j$ along the $j$ segment of the integration path, respectively. $\omega_{j,avg}$ is the weighting averaged wind field along the $j$ segment of the integration path, and $\sigma_{\omega z}$ and $\sigma_{\theta z}$ are the uncertainties in the corrected wind field and wind direction, respectively.

**2.5 Regression model for trend analysis**

We establish a regression model expressed as equations (10) and (11) to simulate the seasonal and inter-annual variabilities of transboundary transport CO flux along the Tibetan Plateau boundary. This model consists of a third-order Fourier series and a linear function. We refer to Sun et al. (2021) for detailed description of its resampling methodology. The fitting process steps are detailed in Figure 1.

$$Y^{org}(t) = Y^{mod}(t) + \varepsilon(t) \tag{10}$$

$$Y^{mod}(t) = A_0 + A_1 t + A_2 \cos\left(\frac{2\pi t}{365}\right) + A_3 \sin\left(\frac{2\pi t}{365}\right) + A_4 \cos\left(\frac{4\pi t}{365}\right) + A_5 \sin\left(\frac{4\pi t}{365}\right) \tag{11}$$

$$d\% = \frac{Y^{org}(t) - Y^{mod}(t)}{Y^{mod}(t)} \times 100 \tag{12}$$

where $Y^{org}(t)$ and $Y^{mod}(t)$ are the original and the fitted time series of CO flux, respectively. $A_0$ is the intercept, $A_1$ is the annual growth rate, and $t$ represents the number of days since May 2018. The $\varepsilon(t)$ is the residual between the original and the fitted results. The coefficients $A_2 - A_5$ describe the seasonal cycle and $A_1/A_0$ is the inter-annual trend. Deviations of CO fluxes from their average seasonal values, known as seasonal enhancements, are calculated according to Equation (12).

**3 Variability of CO and transboundary transport flux**

**3.1 CO Variability over the Tibetan Plateau**

The CO total column over the Tibetan Plateau and its surrounding regions, derived from TROPOMI observations (Fig. S2 and Fig. S3), was analyzed to investigate the diurnal, seasonal, and interannual variations along the Tibetan Plateau boundary. Additionally, fire data from the Moderate Resolution Imaging Spectroradiometer (MODIS) for 2019–2023 (Figs. S5–S9) were incorporated to investigate potential influencing factors. The results show that the daily mean CO total column ranges from $7.75 \times 10^{17}$ molec cm$^{-2}$ to $15.61 \times 10^{17}$ molec cm$^{-2}$, while the seasonal mean ranges from $10.06 \times 10^{17}$ molec cm$^{-2}$ to $12.01 \times 10^{17}$ molec cm$^{-2}$. A severe CO pollution event was identified in the spring of 2021, which may be associated with the post-COVID-19 rebound in emissions(Madineni et al., 2021; Davis et al., 2022). Meanwhile, MODIS active fire counts in 2021 indicate a significant increase in fire activity in northern India, Nepal, and surrounding regions. The combined effect of heightened CO concentrations and intensified fire activity likely contributed to the anomalous enhancement of the CO total column in the

southeastern segment of the Tibetan Plateau boundary from January to June 2021. As shown in Figure 3, high levels of CO total column in all seasons are observed in the southeastern segment of the Tibetan Plateau boundary, which is adjacent to southeastern Asia. We employ the regression model to fit the CO total columns averaged along the closed-loop, the southwestern, and the northeastern segments of the Tibetan Plateau boundary. The fitting results in Figure 4 show that the model is capable of accurately capturing and replicating the seasonal and inter-annual variabilities of CO total column. The correlation coefficient ($r$) achieved are 0.84, 0.90, and 0.75, while the root mean square error ($RMSE$) is $0.54 \times 10^{17}$ molec cm$^{-2}$, $0.41 \times 10^{17}$ molec cm$^{-2}$, and $0.76 \times 10^{17}$ molec cm$^{-2}$ for the closed-loop, the southwestern, and the northeastern segments, respectively. Satellite observations and the fitted results show distinct seasonal characteristics. Elevated levels of CO total column are observed along the closed-loop, northeastern, and southwestern segments from late spring to summer, while lower levels occur during autumn and winter. The closed-loop, southwestern, and northeastern segments all exhibit a bimodal seasonal cycle. The closed-loop and southwestern segment have a pronounced peak in late spring and a minor peak in early autumn, whereas the northeastern segment has a minor peak in late spring and a pronounced peak in early autumn. Notably, significant fluctuations in the CO total column averaged along the closed-loop, southwestern, and northeastern segments occur in autumn and winter, with more substantial fluctuations in autumn than in winter. Over the past six years, the CO total column averaged along the closed-loop, southwestern, and northeastern segments have shown an upward annual trends of $0.95 \times 10^{16}$ molec cm$^{-2}$ yr$^{-1}$, $0.77 \times 10^{16}$ molec cm$^{-2}$ yr$^{-1}$, and $1.10 \times 10^{16}$ molec cm$^{-2}$ yr$^{-1}$, respectively.

It is noteworthy that the seasonal variation of surface CO concentrations monitored by CNEMC in Tibet differs from that of the CO total column. Surface CO concentrations peak in winter and spring, while they are at their lowest in autumn. Among all cities in Tibet, except for Shigatse and Nyingchi, where the surface CO concentrations increase by 0.2 mg m$^{-3}$ yr$^{-1}$, the remaining five cities (Ali, Lhasa, Shannan, Nagqu, and Chamdo) show a decrease of 0.03 to 0.09 mg m$^{-3}$ yr$^{-1}$. Shigatse and Nyingchi are located near regions such as Nepal and Assam in India, where CO concentrations are relatively high, which may explain the observed increase in surface CO levels in these areas.

The seasonal variation patterns of surface CO concentration and total column CO differ due to multiple influencing factors. Surface CO concentration is primarily governed by local emission sources such as combustion and industrial activities, whereas total column CO reflects the overall atmospheric CO distribution, including contributions from long-range transport. During winter and spring, increased combustion from heating and agricultural activities leads to elevated surface CO levels, while lower temperatures and reduced surface emissions in autumn contribute to a decline.

The total column CO over the Tibetan Plateau exhibits peak concentrations in late spring and early autumn, likely driven by monsoon transitions, local air mass convergence, seasonal shifts in atmospheric transport, and fluctuations in biomass burning activity. Overall, the distinct seasonal variation patterns of surface CO and total column CO highlight the complex interplay between local emissions and large-scale atmospheric transport processes.

## 3.3 Variability of transboundary transport CO flux

Figure 5 illustrates the seasonal variations in CO total flux across the grid points along the Tibetan Plateau boundary from May 2018 to April 2024, revealing distinct differences between the southwestern and northeastern segments. The southwestern segment is predominantly characterized by CO influx throughout the year, with the strongest influx occurring in winter, while the northeastern segment primarily experiences CO efflux. These spatial differences are closely tied to geographic factors and reflect the influence of distinct atmospheric circulation systems on the southeastern and northwestern segments of the Tibetan Plateau, leading to divergent seasonal CO exchange patterns.

Building upon this, Figure 6 shows the seasonal cycles of both external influx and internal efflux of CO across the closed-loop, southwestern, and northeastern segments of the Tibetan Plateau. Summary of the corresponding statistics are tabulated in Table 1. The results show that the external influx and internal efflux of CO across the closed-loop, the southwestern, northeastern segments in each year are relatively close and have similar seasonal characteristics. High levels of CO flux occur in late autumn to winter, and low levels occur in summer. In most cases, CO flux maximizes in November, December or January, and minimizes in July or August. The month to month variability during late autumn to winter is greater than that in summer. For the closed-loop of the Tibetan Plateau, the CO external influx varies between 7.90 tonnes $s^{-1}$ - 32.73 tonnes $s^{-1}$ and the internal efflux varies between 7.84 tonnes $s^{-1}$ - 29.61 tonnes $s^{-1}$. In comparison, the external influx in the southwestern segment fluctuates between 3.12 tonnes $s^{-1}$ and 20.89 tonnes $s^{-1}$, while the internal efflux ranges from 1.15 tonnes $s^{-1}$ to 8.86 tonnes $s^{-1}$. In the northeastern segment, the external influx varies between 3.42 tonnes $s^{-1}$ and 11.84 tonnes $s^{-1}$, and the internal efflux spans from 5.47 tonnes $s^{-1}$ to 22.69 tonnes $s^{-1}$ (Table 1).

To quantify these trends. We applied an inter-annual regression model to fit the external influx, internal efflux and net flux of CO averaged along the closed-loop, southwestern and northeastern segments of the Tibetan Plateau. The fitting results in Figure 7 show that the model is capable of accurately capturing and replicating the seasonal and inter-annual variabilities of all kinds of fluxes, yielding high correlation coefficients ($r$) and low root mean square errors ($RMSE$). For the closed-loop CO flux, the external influx and net flux show slight positive trends, with inter-annual growth rates of 2.86 Tg CO $yr^{-1}$ and 4.56 Tg CO $yr^{-1}$, respectively. In contrast, the internal efflux displays a slight negative trend of -1.70 Tg CO $yr^{-1}$. In the southwestern segment, the external influx and internal efflux exhibit similar variabilities, with annual mean values of 11.76 tonnes $s^{-1}$ and 4.41 tonnes $s^{-1}$, respectively. The annual growth rates for external influx and internal efflux are 2..31 Tg CO $yr^{-1}$ and -0.28 Tg CO $yr^{-1}$, respectively. In comparison, the external influx and internal efflux in the northeastern segment exhibit notable differences, with average values of 5.94 tonnes $s^{-1}$ and 13.17 tonnes $s^{-1}$, and annual growth rates of 0.64 Tg CO $yr^{-1}$ and -1.22 Tg CO $yr^{-1}$, respectively. The annual growth rates for net fluxes in the southwestern and northeastern segments are 2.59 Tg CO $yr^{-1}$ and 1.86 Tg CO $yr^{-1}$, respectively.

This suggests that in recent years, the Tibetan Plateau has experienced an increase in external influx, while the internal efflux has slightly decreased. The magnitude of the increase in external influx in the southwestern segment is greater than in the northeastern segment. Conversely, the magnitude of the decrease in internal efflux in the northeastern segment is greater than in the southwestern segment.

### 3.3 Uncertainty of CO flux calculation

From May 2018 to April 2024, the external influx and internal efflux of CO averaged along the Tibetan Plateau were 17.70 tonnes $s^{-1}$ and 17.56 tonnes $s^{-1}$, respectively, resulting in a net influx of 0.13 tonnes $s^{-1}$. These estimates are based on TROPOMI overpasses (13:30 local time (LT)). Extrapolating to a full year, the external influx, internal efflux, and net influx are estimated to be 558.19 Tg CO $yr^{-1}$, 553.77 Tg CO $yr^{-1}$, and 4.10 Tg CO $yr^{-1}$, respectively. These values are comparable to CO emission estimated by Borsdorff et al. (2020) for Mexico and Leguijt et al. (2023) for African cities (Borsdorff et al., 2020; Leguijt et al., 2023).

The complex terrain of Tibetan Plateau, with its significant fluctuations in elevation and ground albedo, coupled with the variable mixture of the Asian monsoon and local valley winds, greatly increases the uncertainty of calculating CO flux over the Tibetan Plateau. TROPOMI-based CO flux calculation over the Tibetan Plateau are influenced by various factors (Shaiganfar et al., 2011; Shaiganfar et al., 2017; Tan et al., 2019; Huang et al., 2020). Here we identify two primary sources of uncertainty: CO total column and the closed-loop wind field.

From May 2018 to April 2024, the daily CO total column over the Tibetan Plateau varied from $4.02 \times 10^{17}$ molec $cm^{-2}$ to $64.43 \times 10^{17}$ molec $cm^{-2}$. The averaged standard deviation varied from $0.16 \times 10^{17}$ molec $cm^{-2}$ to $3.46 \times 10^{17}$ molec $cm^{-2}$, corresponding to error margins from 0.81% to 17.89%. The average error in CO total column along the Tibetan Plateau's closed-loop was 4.27%.

Wind speed and direction uncertainties significantly affect CO flux calculations. The Tibetan Plateau's complex terrain exacerbates wind variability. Using formulas (5) and (6), and (9) in Section 2.3, we calculated the uncertainties from the corrected wind field across 22 levels from 1000 hPa to 50 hPa. The averaged wind speed and direction uncertainties were 3.39 m $s^{-1}$ and 54.55°, respectively. Wind speed uncertainty exhibited clear seasonal fluctuations. In spring (MAM), uncertainties ranged from 0.65 to 0.84 m $s^{-1}$, while summer (JJA) saw a wider range of 0.90 to 1.15 m $s^{-1}$. The lowest uncertainties were observed in autumn (SON), ranging from 0.53 to 0.65 m $s^{-1}$. In winter (DJF), uncertainty values were slightly higher, between 0.65 and 0.71 m $s^{-1}$. Wind direction uncertainty also varied seasonally, with values between 18.10° and 26.41° in spring, increasing to 24.17° to 34.09° in summer. Autumn presented the lowest directional uncertainty, ranging from 16.50° to 20.62°, while winter values were comparable to spring, at 19.07° to 23.91° (Table 2).

The average uncertainties for the corrected wind speed and direction were 0.78 m $s^{-1}$ and 22.15°. In calculation of the closed-loop CO flux, considering only the uncertainty in wind speed results in an average error of 6.99% in CO flux, while accounting solely for wind direction uncertainty leads to an average error of 11.03%. The total error induced by both the wind field and CO total column can be calculated using the error propagation equation. The uncertainty in CO flux caused by these factors amounts to 13.81%.

The uncertainty in CO flux estimation significantly affects the interpretation of cross-border CO exchange. Seasonal variations in the uncertainty of wind speed and direction notably influence CO flux calculations, with higher uncertainties in spring and summer, potentially leading to greater fluctuation in flux estimates. In contrast, uncertainty is lower in autumn, making the results from this season more reliable. Regionally, the southeastern boundary of the Tibetan Plateau experiences more pronounced wind field changes due to the Asian monsoon, leading to higher uncertainty compared to the more stable northwestern region. These uncertainties may also impact the interpretation of key pollution events, such as the significant CO pollution event observed

from January to June 2021. During this period, the uncertainties in wind speed (0.67–1.03 m s⁻¹) and wind direction (18.10°–28.24°) were considerable, meaning the actual CO net input could differ significantly from current estimates. While uncertainties do affect the precise values of CO flux, the overall trend remains robust. Therefore, when analyzing seasonal and regional flux variations, particularly during the monsoon-influenced spring and summer, it is crucial to account for the impact of uncertainty.

## 4 Factors driving the variability of transboundary transport CO flux

### 4.1 Differences between southwestern and northeastern segments

For the closed-loop flux, both external influx and internal efflux exhibit significant fluctuations in winter from 2020 to 2023, followed by a rapid decline. Although the net flux also exhibited considerable fluctuations during this period, its seasonal variability was less pronounced compared to the external influx and internal efflux. The net flux is positive from January to May and in August and December, but turns negative in June, July, and from September to November. Specifically, the external influx in the southwestern segment consistently exceeds the internal efflux, whereas in the northeastern segment, the external influx is lower than the internal efflux. Furthermore, the external influx in the southwestern segment closely resembles that of the internal efflux in the northeastern segment, particularly during summer and early autumn when the rates of decline and increase are most pronounced. Conversely, during winter and early spring, the internal efflux in the southwestern segment aligns closely with the external influx in the northeastern segment. Additionally, in summer, the external influx in the northeastern segment surpasses the internal efflux in the southwestern segment.

In the southwestern segment, a significant increase in flux is noted during autumn, followed by a marked decline in late winter. The net flux variations are more intricate, yet they generally remain positive. The averaged net CO flux is recorded at 7.36 tonnes s⁻¹, with an annual growth rate of 2.59 Tg CO yr⁻¹, typically exhibiting low values during the summer. Notably, from December to June in each year, the net flux experiences three distinct peaks. The first peak is observed during late autumn and winter, attributed to increased burning activities, such as winter heating and agricultural burning, alongside the enhanced transport of strong westerly airflow. These conditions contribute to elevated flux levels. Furthermore, cold winter temperatures inhibit the diffusion and dilution of pollutants, facilitating their accumulation in localized areas (Kunhikrishnan et al., 2004). The second peak occurs in spring, likely due to pollutant resuspension and enhanced photochemical production driven by rising temperatures and atmospheric changes(Assessment, 2004; Hung et al., 2022). The third peak is observed around June, coinciding with the strengthening of the South Asian monsoon. This seasonal shift brings increased moisture and airflow, which aids in diluting pollutants and facilitating their transport over long distances (Yu et al., 2017; Bian et al., 2020; Huang et al., 2023).

In the northeastern segment, the external influx peaks in both summer and winter. In winter, there is a marked increase in external influx, followed by two declines in early and late spring. This behavior is likely influenced by the dynamics of the East Asian monsoon across different seasons and varying intensities of burning activities throughout the year. The internal efflux displays a more complex pattern, featuring three peaks and subsequent declines in winter, March, and June, which correspond to the three peaks of net flux observed in the southwestern segment. Similarly, the net flux variabilities in the northeastern segment are quite intricate. The average net

CO flux is -7.23 tonnes $s^{-1}$, with an annual growth rate of 1.86 Tg CO $yr^{-1}$. The peak and trough values occur around autumn, specifically in late autumn to early winter for peaks and late autumn for troughs. This downward trend mirrors that of the internal efflux. Additionally, we observed that: (1) The CO flux transported through the southwestern segment into the Tibetan Plateau, along with its annual growth rate over the past six years, accounted for approximately 66.44% and 77.61% of the total CO flux within the Tibetan Plateau. This indicates that the external influx to the Tibetan Plateau and its changing trend are predominantly influenced by the southwestern segment. While CO transported through the southwestern segment into the Tibetan Plateau is on the rise, the CO flux transported outward through the northeastern segment is declining. In addition, based on the EDGAR CO emission inventory (https://edgar.jrc.ec.europa.eu/, last access: 31 January 2025) and MODIS fire data (Figs. S5 – S9), several urban hotspots in South Asia were identified. In India, CO emissions in regions such as Assam, Bihar, Delhi, Haryana, Punjab, Uttar Pradesh, and West Bengal are primarily driven by transportation and industrial activities, particularly in large cities and industrial corridors. In Bangladesh, traffic congestion and industrial areas, especially in the capital Dhaka, also significantly contribute to CO emissions. In Pakistan, the Khyber Pakhtunkhwa region exhibits elevated CO emissions, which are closely linked to transportation, industrial activities, and seasonal agricultural biomass burning. During the autumn and winter months, the practice of burning crop residue exacerbates the rise in CO concentrations; (2) Around June each year, we observe minor seasonal peaks in the external influx and net flux across the southwestern segment, as well as in the internal efflux across the northeastern segment. This phenomenon may result from the combined effects of atmospheric circulation and the South Asian summer monsoon mechanism. The atmospheric flow prior to the onset of the South Asian monsoon transports pollutants such as CO from South Asia to the Tibetan Plateau, leading to increased CO concentrations and fluxes in the region (Yu et al., 2017; Huang et al., 2023). This seasonal peak also highlights the intricate interactions between atmospheric circulation and the monsoon system in South Asia during the pollutant transmission process. (3) The southwestern segment exhibits distinct pollutant transport differences compared to the northeastern segment. For example, the averaged external influx across the southwestern segment reaches as high as 11.99 tonnes $s^{-1}$, while the averaged internal efflux across the northeastern segment is only 4.60 tonnes $s^{-1}$. This is primarily attributed to a higher level of rapid industrialization and urbanization in Southeast Asia than in Tibetan Plateau, resulting in higher pollutant emissions (including CO) in the region.

Eastern China is predominantly downwind of the Tibetan Plateau, where the wind flow and atmospheric stability in the upper atmosphere predispose the eastern region to act as a receiving area for pollutants from the Tibetan Plateau. The Tibetan Plateau functions as a high-altitude natural barrier, effectively limiting the in-depth spread of pollutants (Ji et al., 2015). However, the rapid industrialization and urbanization in East China have led to high local pollutant emissions. Influenced by atmospheric circulation patterns and complex topography, East China experiences strong convection and large-scale circulatory systems. While most pollutants are recirculated and deposited within the region, significant transport occurs towards South Korea, Japan, and the North Pacific. A smaller fraction may also reach the eastern edge of the Tibetan Plateau. (Zhang et al., 2015; Yan and Bian, 2015).

**4.2 Spatiotemporal distribution of CO**

We have analyzed spatiotemporal distribution of CO total column from May 2018 to April 2024 over the Tibetan Plateau and its surrounding regions. CO total columns were averaged on both annual and seasonal timescales. Specifically, we estimated the average of CO total column for winter (December to February), the pre-monsoon period (March to May), the monsoon period (June to September), and the post-monsoon period (October to November). Seasonal variations were assessed by subtracting the mean annual CO total column from their seasonal averages. The resulting data are presented in Figure 8. Additionally, we analyzed the six-year average spatial distribution and correlation of CO total column concentrations across the Indian states of Himachal Pradesh, Uttarakhand, and Ladakh, as well as Nepal and Assam. The corresponding results are presented in Figure 9.

The results indicate that CO concentrations over the Tibetan Plateau are consistently lower than those in South Asia throughout the year. CO concentrations in South Asia are particularly elevated, especially during winter and spring, when the disparity is most pronounced. In summer, CO over the Tibetan Plateau disperses across a broader area. Notably, we observed that CO concentrations over the Tibetan Plateau are significantly higher than the annual average during the monsoon season. Specifically, the average CO concentration over the Tibetan Plateau increased by $0.90 \times 10^{17}$ molec cm$^{-2}$ compared to the annual mean, whereas in India and Nepal, CO concentrations decreased by $-2.62 \times 10^{17}$ molec cm$^{-2}$ and $-0.36 \times 10^{17}$ molec cm$^{-2}$, respectively. These findings suggest that CO pollutants from South Asia are transported into the Tibetan Plateau during the South Asian monsoon. Over the past six years, the CO total column in northwestern India, Nepal, and Assam has exhibited relatively high growth rates, with annual increases of $1.55 \times 10^{16}$ molec cm$^{-2}$ yr$^{-1}$, $2.37 \times 10^{16}$ molec cm$^{-2}$ yr$^{-1}$, and $2.29 \times 10^{16}$ molec cm$^{-2}$ yr$^{-1}$, respectively.

The increase of CO over the Tibetan Plateau during summer may be influenced by CO influx driven by the South Asian monsoon, as well as various meteorological factors. High temperatures and intense solar radiation in summer raise the atmospheric mixing layer height, facilitating the easier dispersion of pollutants in the atmosphere (Yang et al., 2004; Huang et al., 2023). Additionally, summer precipitation and wind speed affect the spatial distribution and transport pathways of pollutants. Elevated temperatures and strong convective conditions enhance vertical mixing and horizontal transport, resulting in more extensive and rapid diffusion of pollutants across the Tibetan Plateau (Zhang et al., 2020; Sun et al., 2021).

During winter and spring, CO concentrations rise significantly in South Asian regions, including Nepal, Bhutan, Assam in India, and parts of Myanmar. This increase is likely driven by intensified human activities, such as biomass burning, which is common in these areas during these seasons for heating and agricultural waste disposal, leading to substantial CO emissions. The southern Tibetan region, bordering northern Assam, serves as a key pathway for pollutant transport due to its distinct plain topography. Studies have confirmed that persistent organic pollutants, HCHO, and other contaminants are transported along the Yarlung Tsangpo River valley into the Tibetan Plateau (Sheng et al., 2013; Xu et al., 2024). TROPOMI remote sensing data further show that CO pollutants infiltrate the Tibetan Plateau through this region.

Despite the varying geographical, climatic conditions, and emission sources in these regions, the changes in CO concentrations are interrelated. CO concentrations tend to be lower in high-altitude areas, such as Himachal Pradesh, Uttarakhand, and Ladakh. Nepal, particularly the Kathmandu Valley, faces unique geographic and climatic challenges due to its dense population and concentration of industrial activities, exacerbating air pollution in the region (Islam et al.,

2020). The Kathmandu Valley's encirclement by high mountains makes it especially vulnerable to pollution. In contrast, Assam's plains are heavily influenced by the monsoon and high humidity, which promotes the diffusion and deposition of pollutants. Despite these regional differences, CO concentrations in these areas show strong correlations with data from the Ali region, Nyingchi City, Shannan City, and Shigatse City on the Tibetan Plateau, with correlation coefficients ($r$) ranging from 0.52 to 0.71. These correlations suggest that long-distance pollutant transport, influenced by meteorological conditions such as monsoons, may link these regions to similar or shared pollution sources (Carrico et al., 2003).

The rapid increase in CO concentrations from South Asia, particularly India and Nepal, is closely linked to the rise in flux over the southwestern Tibetan Plateau, driven primarily by industrialization, agricultural activities, and population growth in the region. The topography of the Tibetan Plateau, combined with the monsoon system, facilitates the transport of pollutants from South Asia to the plateau. At the same time, the Tibetan Plateau's natural ability to regulate CO is critical. Furthermore, China's domestic CO emissions have significantly decreased due to policy controls and economic restructuring. The Tibetan Plateau has long been regarded as an atmospheric background, with local anthropogenic emissions deemed negligible(Yao et al., 2012; Zheng et al., 2018; Kang et al., 2019; Sun et al., 2021). Overall, there is a dual trend of increasing external influx and decreasing internal efflux, with the concentration of CO received by the Tibetan Plateau from South Asia exceeding the influence of emissions from inland China.

## 4.3 Transboundary transport pathway

Spatial distribution of CO across four seasons around the Tibetan Plateau is shown in Figure 10, and Figure 11 presents the atmospheric circulation patterns at 200 hPa and 500 hPa, including mean horizontal wind vectors and latitude-height and longitude-height distributions. Significant seasonal variations in CO concentration are observed in the Tibetan Plateau and surrounding areas, primarily influenced by atmospheric circulation patterns, pollutant source strength, and deep convection activities. The interplay among these factors contributes to the complex dynamics of CO distribution, revealing the intricate relationship between local emissions and regional meteorological conditions.

The south Asian summer monsoon transports a substantial amount of air from the surface to the stratosphere, characterized by southwesterly winds in the lower troposphere and an anticyclonic circulation in the upper troposphere (Abe et al., 2013). This anticyclonic system, dominant in the upper troposphere and lower stratosphere, significantly affects CO distribution in the Tibetan Plateau by enhancing the transport of tropospheric pollutants into the stratosphere (Huang et al., 2023). The high-altitude terrain of the Tibetan Plateau amplifies this process, facilitating vertical lifting of air. Surface pollutants are transported to the upper troposphere through the Asian summer monsoon anticyclone and become confined within the South Asian High's anticyclonic system (Randel et al., 2010; Bian et al., 2012; Bian et al., 2020; Huang et al., 2023). This dynamic leads to a significant increase in CO concentration in the upper troposphere and lower stratosphere. During the summer monsoon season, the southwesterly monsoon winds carry substantial pollutants into the plateau, strongly influenced by intense deep convection. These winds uplift CO from the southern plateau and disperse it across the region (Fu et al., 2006). Large-scale deep convection plays a crucial role in lifting CO from upwind source regions to

higher altitudes. While some CO returns to the source region, a portion is transported to the Tibetan Plateau by upper-level southwesterly winds.

In contrast, under dry winter monsoon conditions, CO can be transported to the Himalayan-Tibetan Plateau via the westerlies. The northwesterly flow rapidly conveys CO pollutants from the Northern Hemisphere to the Tibetan Plateau and its surrounding regions (Zhang et al., 2015; Sun et al., 2021). This winter flow typically introduces strong cold air, causing intense surface cooling upon entering the plateau and resulting in descending air currents. This process enhances local circulation, exacerbating CO accumulation (Liu et al., 2003; Zhang et al., 2015). Additionally, the plateau's topographical features influence CO distribution, as stable atmospheric stratification limits vertical dispersion, leading to accumulation in the lower troposphere.

However, the amount of CO transported to the plateau is also influenced by the location and intensity of sources, air mass trajectories, and transport timing (Yao et al., 2012; Zhang et al., 2015; Kang et al., 2019; Sun et al., 2021). Variations in CO concentration depend not only on seasonal atmospheric circulation patterns but also on the distribution and intensity of pollution sources and the frequency and strength of deep convection activity. These complex interactions lead to significant seasonal changes in CO concentrations.

**5 Conclusions**

In this study, we utilized data products of TROPOMI, ERA5, and GEOS-CF, along with the closed-loop integral method, to quantify transboundary transport flux of CO over the Tibetan Plateau. The variabilities and driving forces of external influx, internal efflux, and net flux of CO over the closed-loop, southwestern, and northeastern segments of Tibetan Plateau were analyzed.

The closed-loop CO concentration along the Tibetan Plateau boundary shown significant spatiotemporal variations, with daily means ranging from $7.75 \times 10^{17}$ to $15.61 \times 10^{17}$ molec cm$^{-2}$ and seasonal means from $10.06 \times 10^{17}$ to $12.01 \times 10^{17}$ molec cm$^{-2}$, and with high densities in the southeastern segment adjacent to southeastern Asia. The closed-loop, southwestern, and northeastern segments exhibit a bimodal cycle, peaking in late spring and early autumn, with significant autumn fluctuations and less variability in winter. During the South Asian monsoon, CO concentrations increased by $0.90 \times 10^{17}$ molec cm$^{-2}$ in the Tibetan Plateau, decreased by $1.78 \times 10^{17}$ molec cm$^{-2}$ in India, and decreased by $0.36 \times 10^{17}$ molec cm$^{-2}$ in Nepal compared to the annual average. A strong correlation and synchronization of CO concentrations were observed between the South Asian border region and the Tibetan Plateau. Over the past six years, CO total columns in Tibet, India, and Nepal exhibited growth trends of $0.76 \times 10^{16}$ molec cm$^{-2}$ yr$^{-1}$, $1.20 \times 10^{16}$ molec cm$^{-2}$ yr$^{-1}$, and $1.63 \times 10^{16}$ molec cm$^{-2}$ yr$^{-1}$, respectively. These trends are notably higher than the growth observed in Tibet. Over six years, growth trends of $0.94 \times 10^{16}$ molec cm$^{-2}$ yr$^{-1}$, $0.77 \times 10^{16}$ molec cm$^{-2}$ yr$^{-1}$, and $1.02 \times 10^{15}$ molec cm$^{-2}$ yr$^{-1}$ were observed for each segment.

Transboundary transport flux of CO in the Tibetan Plateau is high in late autumn and winter, and low in summer. Six-year averaged external influx, internal efflux, and net flux are 17.70 tonnes s$^{-1}$ and 17.56 tonnes s$^{-1}$, 0.13 tonnes s$^{-1}$, respectively. The external influx shows a slight positive trend of 2.86 Tg CO yr$^{-1}$, while net flux increases at -4.56 Tg CO yr$^{-1}$, contrasted by a minor decline in internal efflux of -1.70 Tg CO yr$^{-1}$. In the southwestern segment, external influx and internal efflux show comparable variability, with annual means of 11.76 tonnes s$^{-1}$ and 4.41 tonnes s$^{-1}$, and growth rates of 2.31 Tg CO yr$^{-1}$ and -0.28 Tg CO yr$^{-1}$, respectively. Conversely, the

northeastern segment exhibits significant differences, with average influx and efflux of 5.94 tonnes s$^{-1}$ and 13.17 tonnes s$^{-1}$, and growth rates of 0.64 Tg CO yr$^{-1}$ and -1.22 Tg CO yr$^{-1}$, respectively. In summary, these trends indicate an increase in external influx and a slight decrease in internal efflux across the Tibetan Plateau, with significant regional differences in CO fluxes; the southwestern segments serves as the primary contributor to external influx, exhibiting considerable seasonal changes, while the eastern segments shows lower external influx than internal efflux, indicating a net efflux.

We assessed the uncertainties of wind speed and direction across 22 layers from 1000 hPa to 50 hPa, obtaining average uncertainties of 3.39 m s$^{-1}$ for wind speed and 54.55° for wind direction, with corrected averages of 0.78 m s$^{-1}$ and 22.15°, respectively. The uncertainty in wind speed accounts for an average error of 6.99%, while the uncertainty in wind direction contributes an average error of 11.03%. The average error in CO total column along the Tibetan Plateau's closed-loop was 4.27%. Using the error propagation equation, the total uncertainty in CO flux from both factors is calculated to be 13.81%.

In conclusion, we quantified the CO flux over the Tibetan Plateau and found a significant seasonal trend, with an increasing external influx in recent years. Specifically, the southwestern segments of the Tibetan Plateau represent the primary source of CO, demonstrating an upward trend potentially associated with the rapid increase in CO concentrations from South Asia. Conversely, CO transmission to the eastern segments is declining, likely due to decreased emissions and the plateau's natural ability to regulate CO. The unique geographical position of the Tibetan Plateau makes it crucial for observing and investigating transboundary transport of regional atmospheric pollutants, providing a scientific basis for understanding global pollutant transport mechanisms and informing environmental protection policies.

This study faces some limitations due to the lack of cross-validation with ground-based measurements. However, datasets such as TROPOMI, ERA5, and GEOS-CF, which are generally reliable, provide a solid foundation for the research. To address these limitations, future research will incorporate both ground-based remote sensing and in situ measurements for validation. Furthermore, the ongoing third scientific expedition to the Tibetan Plateau, led by the Chinese government, is expected to improve data availability, offering valuable opportunities to refine our findings.

*Data availability.* The TROPOMI CO dataset of this study is available for download at https://scihub.copernicus.eu/ (last access: 16 June 2024). ERA5 hourly wind data are available download at https://cds.climate.copernicus.eu/ (last accessed: 1 June 2024). GEOS-CF simulations are available for download at https://gmao.gsfc.nasa.gov/ (last accessed: 12 April 2024). The EDGAR CO emission inventory is available at https://edgar.jrc.ec.europa.eu/ (last access: 31 January 2025). GEOS-Chem simulations in this study are available on request from Youwen Sun (ywsun@aiofm.ac.cn)

*Author contributions.* ZS carried out the data analysis and prepared the paper with input from all coauthors. YS designed the study. HY conducted the GEOS-Chem simulations and, along with XL, YS, ZP, CYL, and CL, provided constructive comments. HY also offered valuable insights into the data analysis.

*Competing interests.* The authors declare that they have no conflict of interest.

*Acknowledgements.* We thank the NASA Global Modeling and Assimilation Office (GMAO) for providing the GEOS-CF simulations, and the Copernicus Climate Change Service (C3S) for providing the ERA5 reanalysis data. We also express our gratitude to the GEOS-Chem team for sharing the model, and NOAA for providing the GEOS-FP meteorological files. Additionally, we thank the European Space Agency (ESA) for providing the Sentinel-5P TROPOMI CO data.

*Financial support.* This work is jointly supported by the National Science Fund for Excellent Young Scholars (No. 62322514, 62322513), National Science Foundation of China (12174367), Anhui Science Fund for Distinguished Young Scholars (No. 2308085J25) and National Key Research and Development Program of China (No. 2023YFC3709502).

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

**Figures**

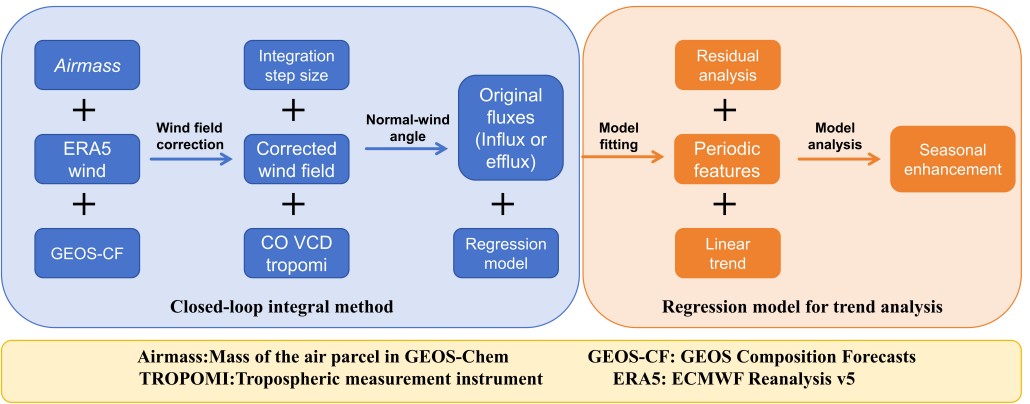

**Fig.1.** Schematic illustrating the closed-loop integral method for calculating input-output fluxes using different data products, and the regression model fitting method. The different data products, closed-loop integral method, and regression model are discussed in Sections 2.2-2.6, respectively.

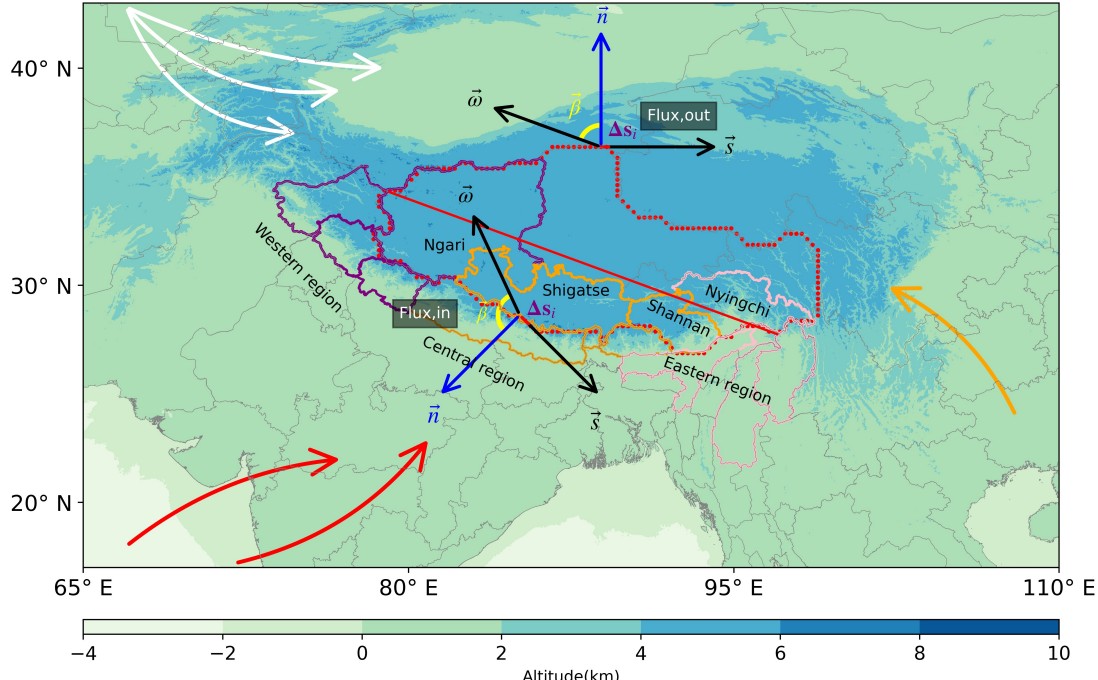

**Fig. 2.** Geographical description of the Tibetan Plateau and demonstration of the closed-loop integral method for

CO flux calculation, with red points indicating the closed-loop. The black arrow $\boldsymbol{\omega}$ represents the wind field vector,

$\boldsymbol{s}$ is the step vector for integration, and the blue arrow $\boldsymbol{n}$ denotes the boundary normal vector. $\boldsymbol{\beta}$ is the angle between

the wind field vector $\boldsymbol{\omega}$ and the boundary normal vector $\boldsymbol{n}$, while $\Delta\boldsymbol{s}$ represents the integration step length. White

arrows indicate the westerly circulation over the plateau, red arrows represent the South Asian monsoon, and

orange arrows indicate the East Asian monsoon. Surrounding areas are categorized into western (in purple), central

(in orange), and eastern (in pink) regions. Within the Tibetan Plateau, highlighted areas include Ngari Prefecture,

Shigatse City, Shannan City, and Nyingchi City. Elevation data is sourced from NOAA NCEI. The red bold line

split the closed-loop of the Tibetan Plateau into southeastern and northeastern segments.

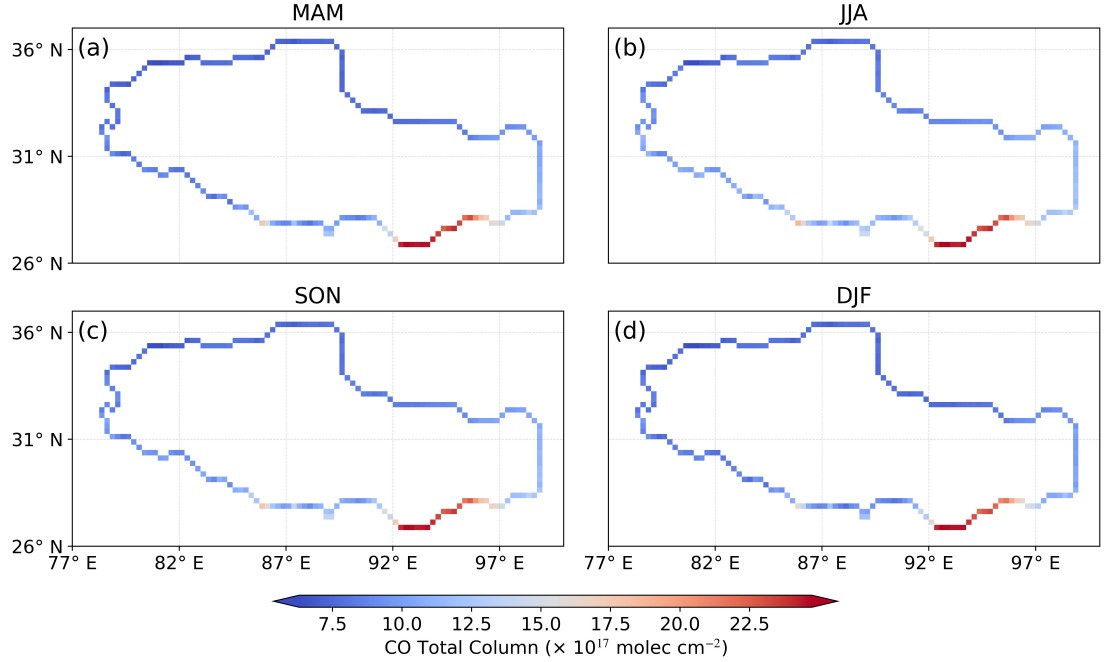

**Fig. 3.** Seasonal average of CO total column over the Tibetan Plateau, which is derived from data collected across

1    all days from May 2018 to April 2024, and is categorized by spring, summer, autumn, and winter.

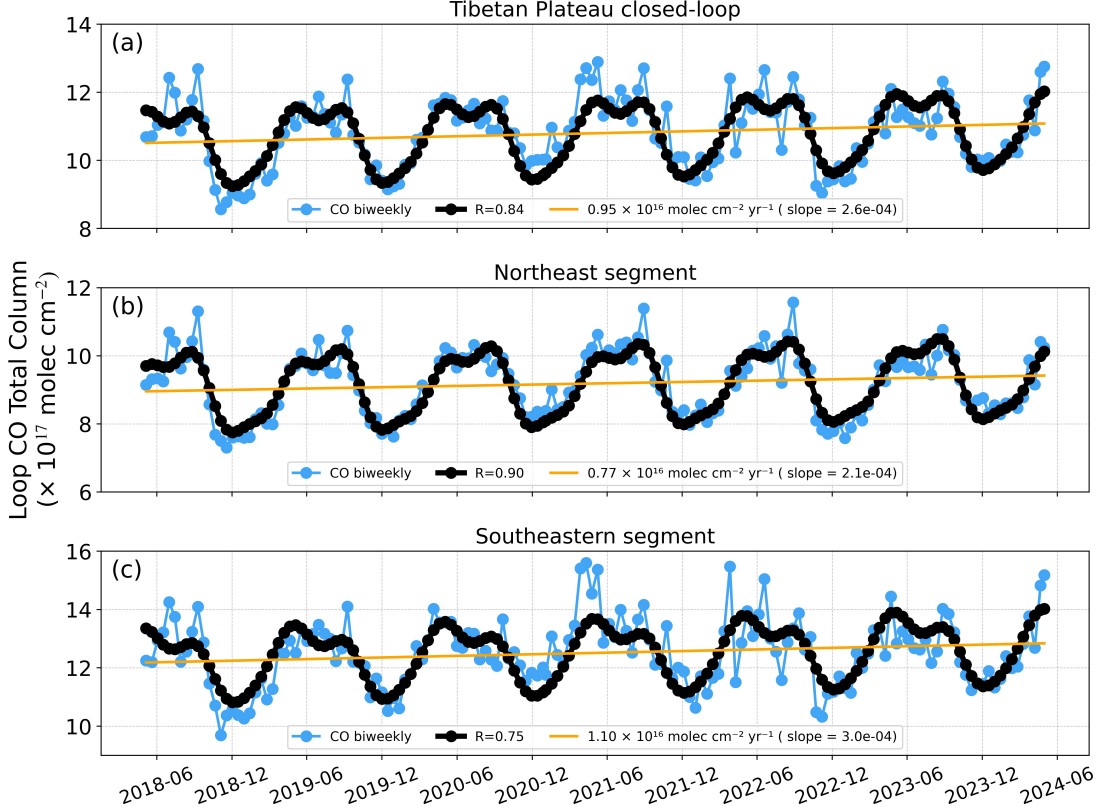

**Fig. 4.** Panels (a), (b), and (c) depict the inter-annual variabilities of CO concentrations in the closed-loop of the

Tibetan Plateau, as well as the northeastern and southwestern segments, from May 2018 to April 2024. Blue dots

represent biweekly averaged CO total column. The figures illustrate the seasonal trend (black line) and

inter-annual trend (orange line) fitted by the seasonal cycle model.

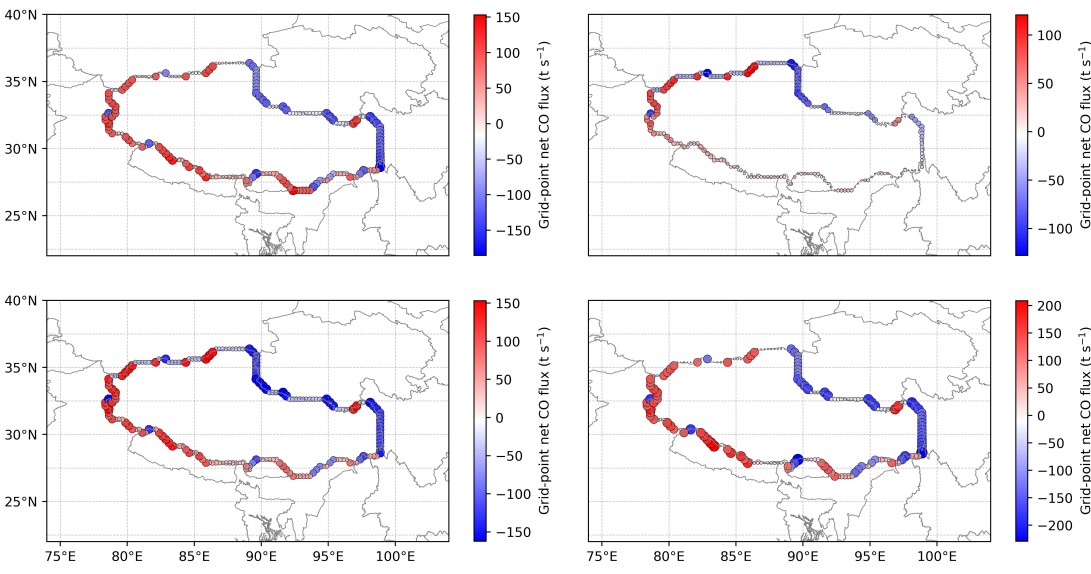

**Fig. 5.** The total CO flux for each grid across different seasons, based on data from May 2018 to April 2024,

categorized into spring, summer, autumn, and winter. Red indicates external influx to the Tibetan Plateau, while

blue represents internal efflux. Bubble size is proportional to flux magnitude, with positive and negative values

denoting influx and efflux, respectively.

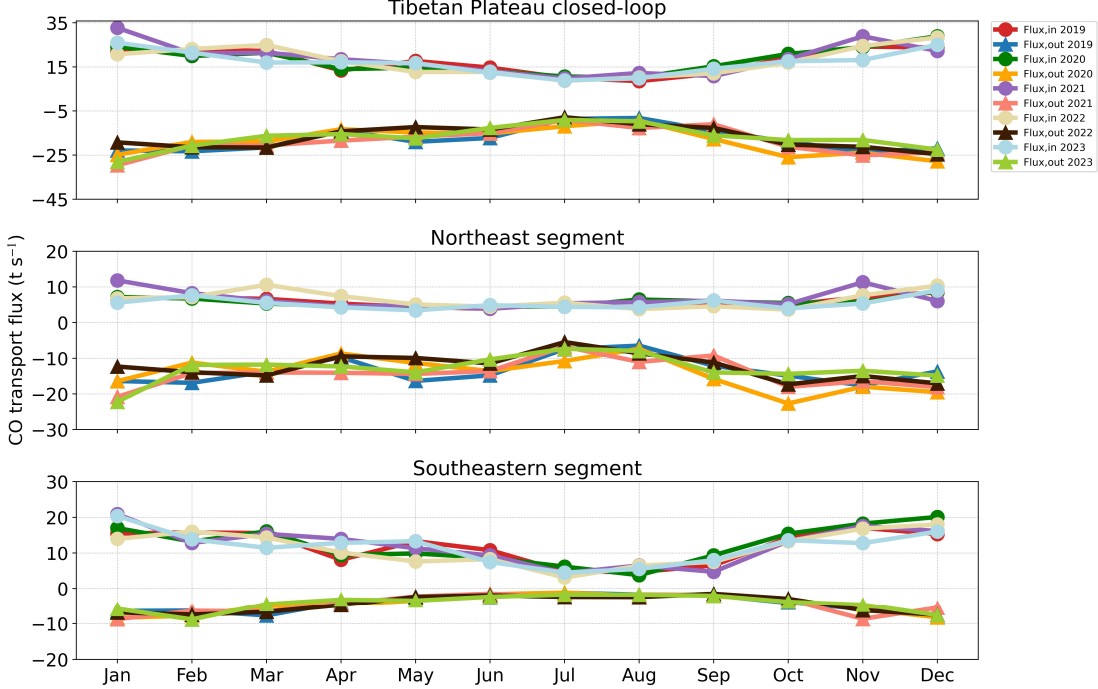

**Fig. 6.** Monthly averaged external influx and internal efflux of CO over the closed-loop, the southwestern, and the northeastern segments of the Tibetan Plateau. Results are presented based on five complete years (2019–2023).

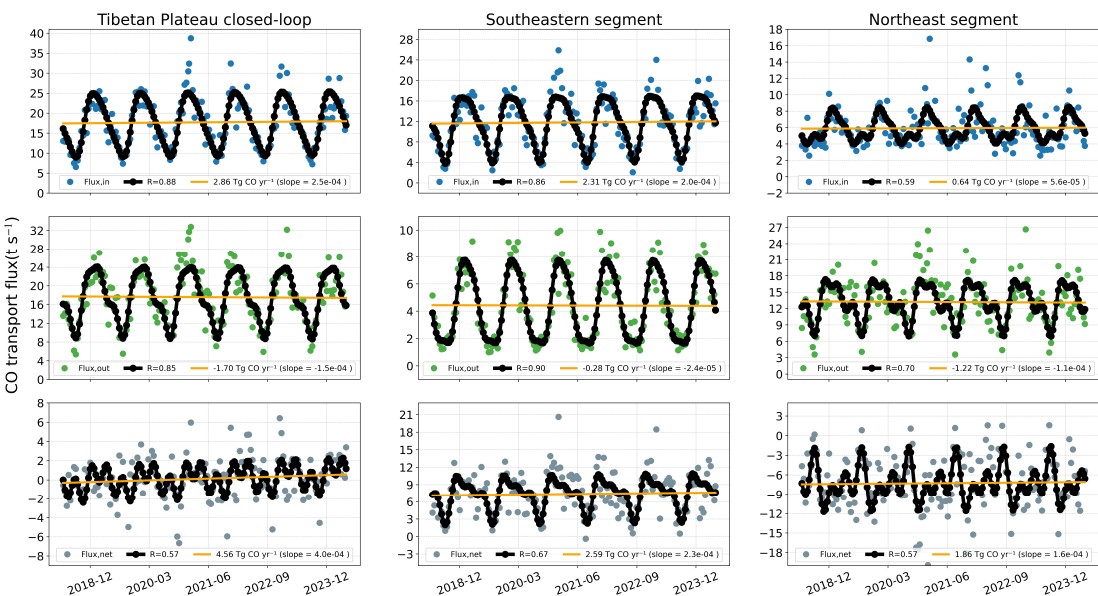

**Fig. 7.** Inter-annual variabilities of external influx, internal efflux and net flux of CO over the closed-loop, the southwestern, and the northeastern segments of the Tibetan Plateau from May 2018 to April 2024. Blue dots represent external influx, green dots indicate internal efflux, and gray dots show net flux. The seasonal trend (black line) and inter-annual trend (orange line) are fitted using the seasonal cycle model.

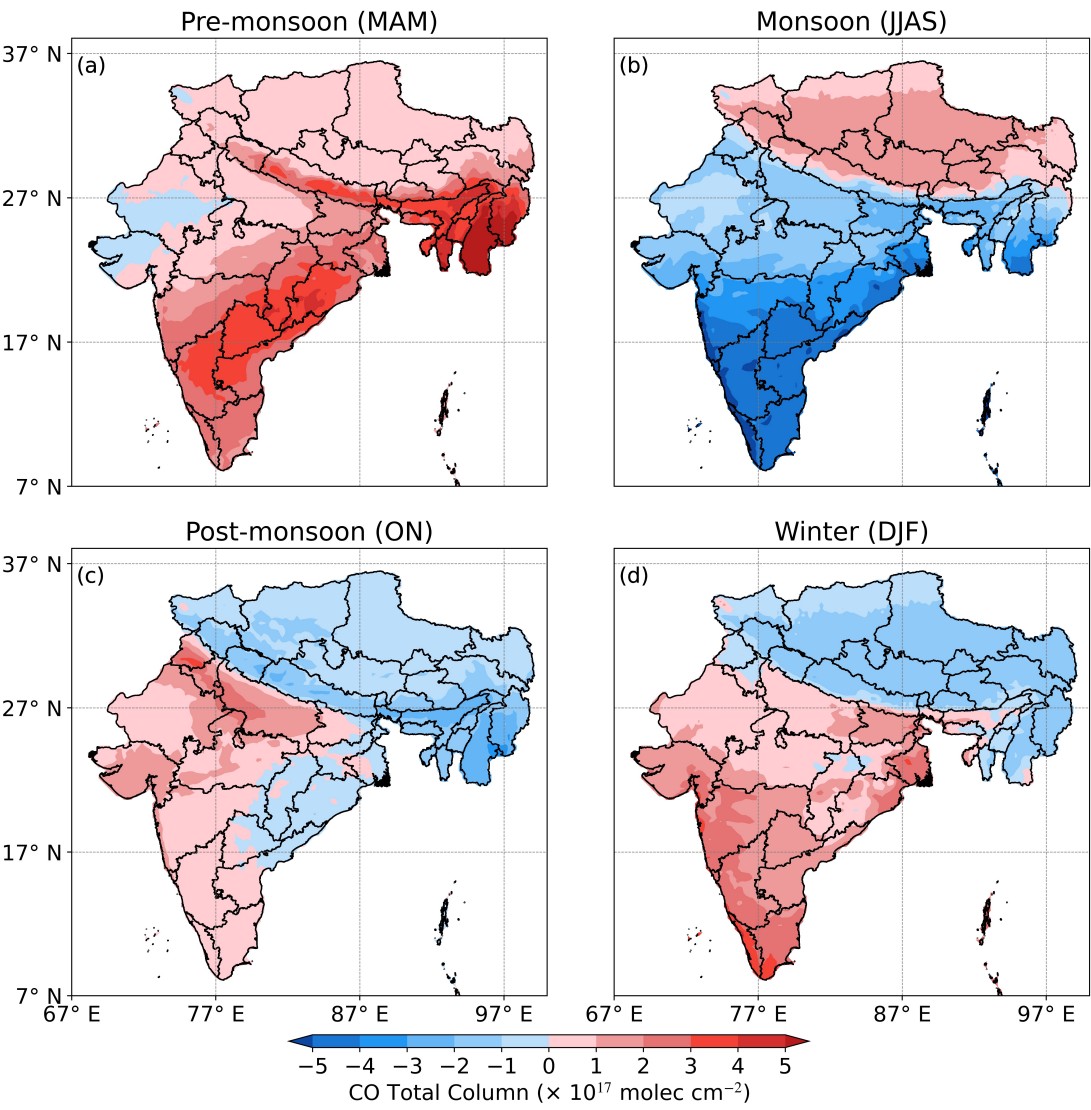

**Fig. 8.** The seasonal averages of CO columns over the Tibetan Plateau and South Asia during the pre-monsoon (MAM), monsoon (JJAS), post-monsoon (ON), and winter (DJF) periods have been adjusted by removing the annual mean CO columns. The data were collected from May 2018 to April 2024.

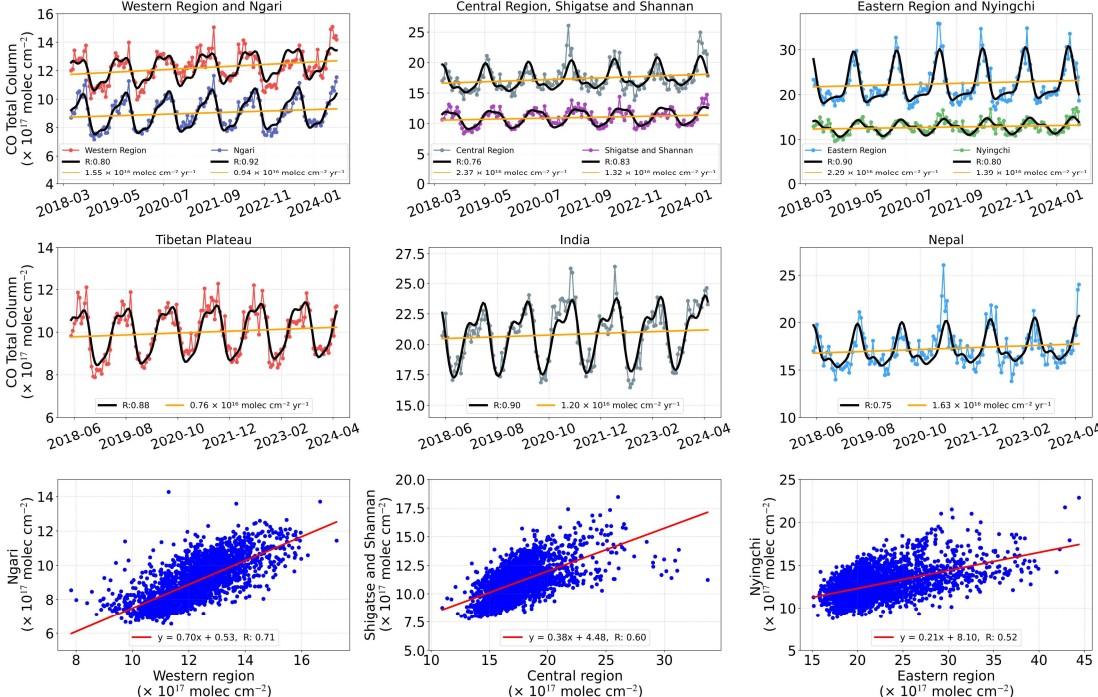

**Fig. 9.** Seasonal, inter-annual variabilities, and correlation analysis of CO concentrations over the western, central, and eastern regions outside the Tibetan Plateau, and over the Ngari, Shigatse, Shannan, and Nyingchi within the Tibetan Plateau. The red, dark blue, gray, purple, blue, and green dots in the figure represent CO concentrations in the western region outside the Tibetan Plateau, Ngari, the central region outside the Tibetan Plateau, Shigatse, Shannan, the eastern region outside the Tibetan Plateau, and Nyingchi, respectively.

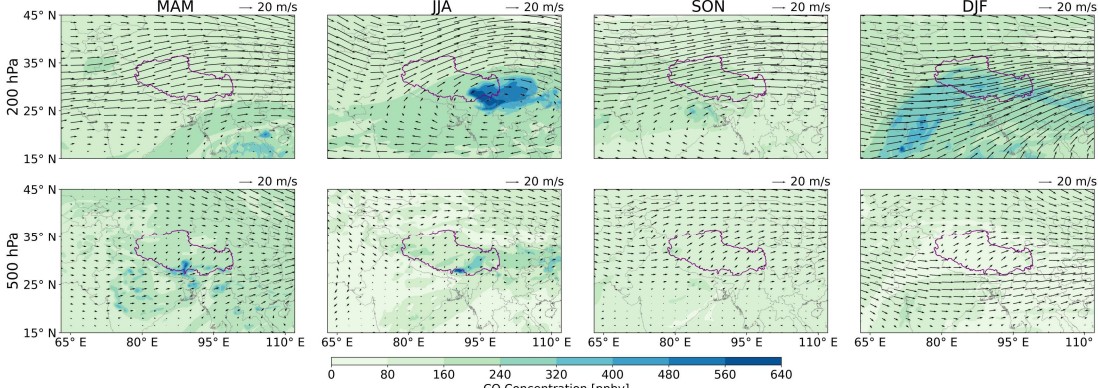

**Fig. 10.** Spatial distribution of CO concentration surrounding the Tibetan Plateau across different seasons, alongside mean horizontal wind vectors at 200 hPa and 500 hPa, represented by arrows. The study area is outlined in purple. The CO spatial distribution data is available from GEOS-CF, while the meteorological fields are derived from ERA5.

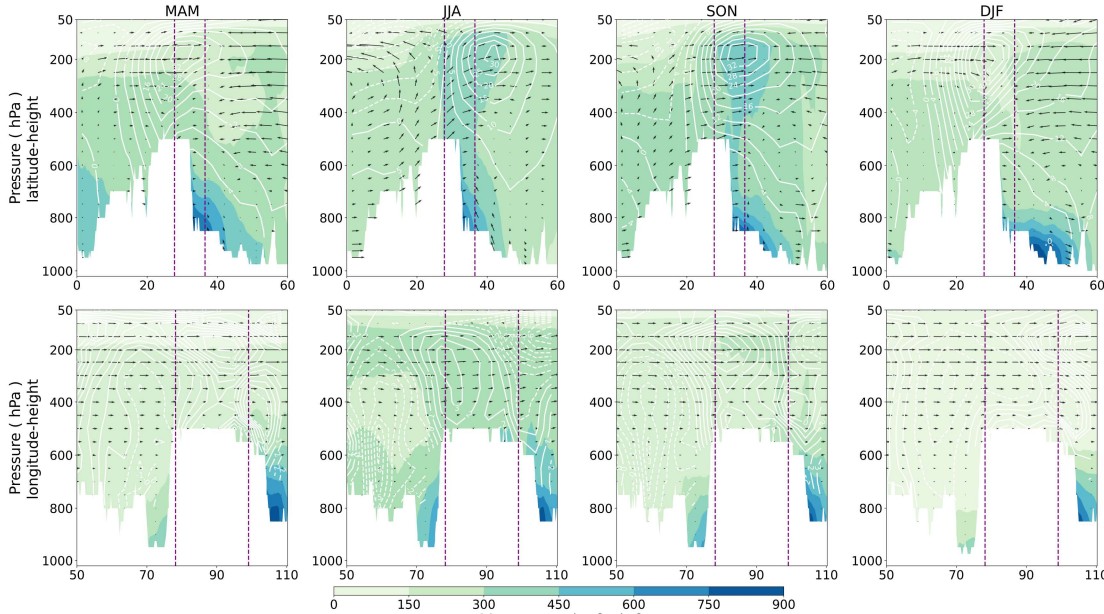

Fig. 11. The first row shows the latitudinal–altitude distribution of CO concentrations in different seasons averaged over the range 50°–110° E (positions correspond to different columns). The white contours at intervals of 18 m s$^{-1}$ represent the westerly (solid) and easterly (dashed) mean meridional winds; the white areas represent the terrain, and the arrows represent the wind vectors (vertical speed units are $10^{-4}$ hPa s$^{-1}$, zonal wind units are m s$^{-1}$); the study area is marked by the purple dashed line. The second row is calculated from the longitude–altitude angles, averaged over the range 27°–33°N. Here the white contours represent the southerly (solid) and northerly (dashed) mean zonal winds, and the horizontal component of the wind vector is the meridional wind (m s$^{-1}$). The meteorological fields are from ERA5.

# Tables

**Table 1**. Statistics of the external influx and internal efflux of CO across the closed-loop, the southwestern and northeastern segments of the Tibetan Plateau from May 2018 to April 2024.

| Year / Type | | | 2018 | 2019 | 2020 | 2021 | 2022 | 2023 | 2024 |
|---|---|---|---|---|---|---|---|---|---|
| Seasonal cycle (monthly mean) | Tibetan | Flux,in max/min (t s⁻¹) | 22.69/7.90 (Dec./Aug.) | 24.21/8.54 (Nov./Aug.) | 28.88/10.24 (Dec./Aug.) | 32.73/9.77 (Jan./Jul.) | 28.33/8.69 (Dec./Jul.) | 25.95/8.88 (Jan./Jul.) | 23.12/18.40 (Mar./Apr.) |
| | | Flux,out max/min (t s⁻¹) | 22.45/7.89 (Dec./Aug.) | 23.56/8.16 (Nov./Aug.) | 27.80/9.37 (Dec./Aug.) | 29.61/9.10 (Jan./Jul.) | 24.61/7.84 (Dec./Jul.) | 28.08/9.05 (Jan./Jul.) | 22.18/16.65 (Mar./Jan.) |
| | southwestern segment | Flux,in max/min (t s⁻¹) | 15.21/3.57 (Dec./Aug.) | 17.04/4.66 (Nov./Aug.) | 20.09/3.77 (Dec./Aug.) | 20.89/4.35 (Jan./Jul.) | 17.98/3.12 (Dec./Jul.) | 20.36/4.48 (Jan./Jul.) | 16.69/11.10 (Mar./Jan.) |
| | | Flux,out max/min (t s⁻¹) | 7.18/1.25 (Nov./Jul.) | 8.21/1.19 (Dec./Jul.) | 8.41/1.15 (Jan./Jul.) | 8.79/1.47 (Jan./Jun.) | 7.56/1.49 (Dec./Sep.) | 8.86/1.71 (Feb./Jul.) | 7.14/5.89 (Jan./Mar.) |
| | northeastern segment | Flux,in max/min (t s⁻¹) | 7.89/3.71 (Nov./Oct.) | 8.48/3.88 (Dec./Jun.) | 8.79/4.46 (Dec./Jul.) | 11.84/3.85 (Jan./Jun.) | 10.59/3.63 (Mar./Oct.) | 9.01/3.42 (Dec./May) | 6.80/4.33 (Jan./Apr.) |
| | | Flux,out max/min (t s⁻¹) | 15.97/5.84 (Dec./Aug.) | 17.63/6.47 (Nov./Aug.) | 22.69/7.35 (Oct./Aug.) | 20.82/6.73 (Jan./Jul.) | 17.34/5.47 (Oct./Jul.) | 22.21/7.34 (Jan./Jul.) | 16.30/9.52 (Mar./Jan.) |

**Table 2**. Uncertainties in the corrected mean wind speed and direction for the wind field used for calculating the closed-loop flux of CO over the Tibetan Plateau.

| Year | Averaged Wind Speed Uncertainty (m/s) | | | | Averaged Wind Direction and Its Uncertainty (°) | | | |
|---|---|---|---|---|---|---|---|---|
| | MAM | JJA | SON | DJF | MAM | JJA | SON | DJF |
| 2018 | 0.78 | 1.15 | 0.60 | 0.67 | 22.31 | 34.09 | 18.10 | 19.07 |
| 2019 | 0.65 | 0.92 | 0.60 | 0.71 | 19.84 | 24.90 | 18.23 | 23.91 |
| 2020 | 0.75 | 0.93 | 0.53 | 0.65 | 23.19 | 24.17 | 13.99 | 22.69 |
| 2021 | 0.70 | 1.03 | 0.65 | 0.67 | 18.10 | 28.24 | 20.62 | 21.23 |
| 2022 | 0.84 | 0.90 | 0.59 | 0.68 | 26.41 | 28.05 | 16.50 | 20.02 |
| 2023 | 0.67 | 1.07 | 0.65 | 0.68 | 18.65 | 31.01 | 19.53 | 22.28 |
| 2024 | 0.66 | - | - | - | 19.41 | - | - | - |