# Peer review of "Quantifying transboundary transport flux of CO over the Tibetan Plateau: variabilities and drivers"

_EGUsphere, 2024_

## Author Response (AR1)

**Point-by-point response letter**

**Note: The black font are comments from the referees, and the red font are authors' responses as well as the related change clarifications.**

**(1) Detailed response to comments from referee #1:**

This paper discusses the CO flux and concentration over the Tibetan Plateau. The general idea and research findings are interesting, and I have a few comments to help improve the manuscript:

Specific Comments:

Figure 1: The figure does not clearly explain what the black, blue, orange, and red arrows represent. Please clarify in the caption.

**Response:** Thank you for your comment. We have adjusted the colors to better differentiate what each arrow represents and have updated the figure caption for clarity in the revised version: Figure 2: The black arrow $\omega$ represents the wind field vector, $s$ is the step vector for integration, and the blue arrow $n$ denotes the boundary normal vector. $\beta$ is the angle between the wind field vector $\omega$ and the boundary normal vector $n$, while $\Delta s$ represents the integration step length. White arrows indicate the westerly circulation over the plateau, red arrows represent the South Asian monsoon, and orange arrows indicate the East Asian monsoon.

Figure 3: Why does the southeastern segment show high peaks (blue dots) from January to June 2021? Is this high concentration caused by emissions from South Asia? Please provide an explanation.

**Response:** The high peaks (blue dots) observed in the southeastern segment from January to June 2021 are primarily attributed to increased emissions from South Asia, including Nepal, India, and surrounding regions. A comprehensive analysis of TROPOMI data (Fig. S2 and Fig. S3) reveals a significant rise in CO concentrations during the winter of 2020 (January and February) and the spring of 2021, closely linked to a substantial increase in emissions across this region in early 2021. Additionally, analysis of MODIS fire data (Figs. S5 - S9) indicates a notable rise in fire occurrences in certain parts of South Asia during this period(Govardhan et al., 2023). Multiple studies and reports(Madineni et al., 2021; Davis et al., 2022) suggest that the post-pandemic economic recovery and rising energy demand were key drivers of the sharp rebound in global and South Asian emissions in 2021. In particular, increased energy consumption in the power and transportation sectors contributed to higher emissions.

Davis, S. J., Liu, Z., Deng, Z., Zhu, B., Ke, P., Sun, T., Guo, R., Hong, C., Zheng, B., Wang, Y.,

Boucher, O., Gentine, P., and Ciais, P.: Emissions rebound from the COVID-19 pandemic, Nature Climate Change, 12, 412-414, 10.1038/s41558-022-01332-6, 2022.

Govardhan, G., Ambulkar, R., Kulkarni, S., Vishnoi, A., Yadav, P., Choudhury, B. A., Khare, M., and Ghude, S. D.: Stubble-burning activities in north-western India in 2021: Contribution to air pollution in Delhi, Heliyon, 9, e16939, 10.1016/j.heliyon.2023.e16939, 2023.

Madineni, V. R., Dasari, H. P., Karumuri, R., Viswanadhapalli, Y., Perumal, P., and Hoteit, I.: Natural processes dominate the pollution levels during COVID-19 lockdown over India, Sci Rep, 11, 15110, 10.1038/s41598-021-94373-4, 2021.

Zhang, L., Liu, X., and Wang, Y. (2021). Carbon emissions rapidly rebounded following COVID pandemic dip. Nature, [Online] Available from:
https://www.nature.com/articles/d41586-021-03036-x (Accessed: 24 February 2025).

Figure 5: Consider adding a slope to the yellow line to highlight the decline or increase in flux (also Figure 2) . Additionally, the statement on Page 9, Line 40, "the internal efflux transported through the northeastern segment is declining," is not clearly reflected in Figure 5. Please ensure the figure supports this observation.

**Response:** We have added a slope annotation to the yellow line to more clearly highlight the trend in flux changes. Additionally, we have updated the units for annual growth to ensure a more accurate representation of data variations and to improve the readability and consistency of the figure.We apologize for the confusion caused by our wording, which led to some misinterpretation. What we intended to convey is that "the CO flux transported outward through the northeastern segment is declining", which corresponds to the "flux, out" shown in Figure 7 under the "northeastern segment" title. Additionally, we have revised the original statement "the internal efflux transported through the northeastern segment is experiencing a decline" to "the efflux transported from the Tibetan Plateau through the northeastern segment is declining" for clarity.

Figure 6: The caption does not clearly state that the total column has already been adjusted by subtracting the mean annual CO total column. Please revise the caption for clarity. Can the author also provide the months for those four panels?

**Response:** We have revised the caption for clarity, as requested. The total column data has been adjusted by subtracting the mean annual CO total column. Additionally, we have provided the specific months for the four panels in the revised version of the figure caption.

Page 7, Lines 36–40: Does the flux change lead to an increase in the CO total column over the Tibetan Plateau? Additionally, how does the local CO emission rate change during this period? Please elaborate.

**Response:** The CO concentrations measured by CNEMC and TROPOMI reflect variations in near-surface CO levels and total atmospheric CO column over the Tibetan Plateau, respectively. According to TROPOMI data (Fig. S2 and Fig. S3) and model fitting results (Fig. 9), the total CO column over the Tibetan Plateau increased at a rate $0.76 \times 10^{16}$ molec cm$^{-2}$ yr$^{-1}$ during the study period (Please see Section 4.2

for details). However, the trend in surface CO concentrations observed by CNEMC contrasts sharply with that of the TROPOMI total CO column. With the exception of Shigatse and Nyingchi, surface CO concentrations in most cities exhibited a declining trend. Notably, Shigatse and Nyingchi, located near the borders of Assam, India, and Nepal, experienced an increase in surface CO concentrations at a rate of 0.02 mg m$^{-3}$ yr$^{-1}$ (Please see Section 3.1 for details), likely influenced by emissions from neighboring regions.

Figures 8: Could you add a wind speed legend for the arrows? This would help readers better understand the wind patterns and their impact on CO distribution.
**Response:** We have added the wind speed legend for the arrows in the revised version.

**(2) Detailed response to comments from referee #3:**

The article "Quantifying transboundary transport flux of CO over the Tibetan Plateau: variabilities and drivers," addresses a critical topic with significant implications for atmospheric science and environmental policy. The study presents a comprehensive dataset and employs advanced methods like the closed-loop integral approach and regression modelling to analyse CO transport dynamics over the Tibetan Plateau. By exploring the seasonal characteristics, trends, and source attribution of CO transport, the study aims to contribute to a deeper understanding of the environmental processes shaping the Tibetan Plateau's atmospheric composition. However, there are notable shortcomings that diminish the overall impact and clarity of the research. Below are the key areas for improvement:

General comments:

The study heavily relies on TROPOMI, ERA5, and GEOS-CF datasets. While these are robust, the absence of cross-validation with ground-based measurements reduces the robustness of the conclusions.

Recommendation: It would be better to incorporate ground-based CO measurements or at least discuss the absence of such data as a limitation. Highlight any plans for future validation efforts.

Figures are referenced, but the level of detail provided about them in the main text is limited. For example, geographical disparities between the southwestern and northeastern segments are crucial but are not visually emphasized with appropriate maps or contrasting data visualizations.

Recommendation: Improve figure annotations and provide contrasting visualizations (e.g., heat maps or flow diagrams) to emphasize spatial and seasonal differences in CO flux.

**Response:** In the revised manuscript, we incorporated ground-based CO measurements from the CNEMC monitoring sites to enrich our understanding of surface CO variations on the Tibetan Plateau (Fig. S1), complementing the current analysis of CO in this study. However, due to the unique environmental conditions of the region and the limited ground-based measurement network, obtaining extensive ground data presents a significant challenge. We discuss this limitation in the revised

manuscript and outline plans for future validation efforts.Additionally, we have included a bubble heat maps (Fig. 5) to enhance data visualization and highlight the spatial and seasonal variations in CO flux. Please refer to Sections 3.1, 3.3, and 5 for details.

Section specific comments:

Section 2.3: The closed loop integral method for CO flux calculation

While the closed-loop integral method and regression models are sophisticated, their explanation in the paper is overly technical and lacks sufficient simplification for accessibility. This hinders readers from other disciplines from comprehending the approach. A clearer visual explanation or step-by-step breakdown would enhance understanding.

Recommendation: Including a flowchart or stepwise illustration of the methodology could improve the understanding.

**Response:** In the revised manuscript, we have added a step-by-step flowchart (Fig. 5) to illustrate the closed-loop integral method and regression models, which has helped improve readers' understanding of these methods.

Section 3.1: Variability of CO total column

The findings highlight bimodal seasonal cycles and trends across the Plateau. However, these are repeated across sections without advancing the narrative or exploring less-obvious phenomena like anomalies or outliers.

Recommendation: Identify and explain unusual patterns or deviations from expected trends.

**Response:** In the revised manuscript, we have explored the unusual patterns and outliers, and analyzed the potential factors contributing to these phenomena. Please see section 3.1 for details.

Section 3.3: Uncertainty of CO flux calculation

The uncertainty analysis section provides quantitative insights yet inadequately connects to the study's primary findings. It would be better to explain how these uncertainties impact seasonal and regional trends.

**Response:** In the revised manuscript, we have expanded the uncertainty analysis to include a discussion on how these uncertainties impact seasonal and regional trends. Please see section 3.3 for details.

Section 4.1: Differences between Southwestern and northeastern segments

The paper mentions correlations between South Asia and the Tibetan Plateau, but it misses a deeper quantitative analysis of the mechanisms linking industrial or agricultural activities to observed flux trends. The analysis identifies South Asia as a key CO contributor but lacks granularity regarding specific industrial or agricultural hotspots. For example, no specific industrial hotspots are identified as primary sources, leaving the findings somewhat generic.

Recommendation: Use additional tools (e.g., emission inventories or regional

modelling) to pinpoint primary CO sources. Link trends to specific regions or activities, such as urban centres or biomass-burning zones.

**Response:** We have accepted the reviewer's suggestion and incorporated the EDGAR emission inventory (Fig. S4) and MODIS fire data (Figs. S5 – S9) in the revised manuscript to further analyze emission hotspots and their potential contributions to CO fluxes. We have preliminarily identified industrial centers and urban hotspots to enhance the regional specificity of our results. Please see section 4.1 for details.